# What Is the Best Treatment Choice for Concomitant Ipsilateral Femoral Neck and Intertrochanteric Fracture? A Retrospective Comparative Analysis of 115 Consecutive Patients

**DOI:** 10.3390/jpm12111908

**Published:** 2022-11-16

**Authors:** Han Soul Kim, Dong Keun Lee, Ki Uk Mun, Dou Hyun Moon, Chul-Ho Kim

**Affiliations:** 1Department of Orthopedic Surgery, Gachon University Gil Medical Center, Namdong-gu, Incheon 21556, Republic of Korea; 2Department of Orthopedic Surgery, Chung-Ang University Hospital, Chung-Ang University College of Medicine, Seoul 06973, Republic of Korea

**Keywords:** femoral neck fracture, intertrochanteric fracture, cephalomedullary nail, dynamic hip screw, arthroplasty

## Abstract

Purpose: Although a concomitant ipsilateral femoral neck and intertrochanteric fracture has been considered to be a rare type of injury, its incidence has been increasing, especially among elderly hip fracture patients. However, there is limited evidence on the optimal treatment option. This study surveys surgical outcomes of different implants in order to assist in selecting the best possible implant for a combined femoral neck and intertrochanteric fracture. Methods: The postoperative complications after the treatment of a concomitant ipsilateral femoral neck and intertrochanteric fracture via cephalomedullary nail (CMN), dynamic hip screw (DHS), and hip arthroplasty groups were analyzed by retrospectively reviewing the electronic medical records of 115 consecutive patients. Results: The patient demographics and perioperative details showed no significant discrepancies amongst different surgical groups, except for the operative time; a CMN had the shortest mean operative time (standard deviation) of 85.6 min (31.1), followed by 94.7 min (22.3) during a DHS, and 107.3 min (37.2) during an HR (*p* = 0.021). Of the 84 osteosynthesis patients, 77 (91.7%) achieved a fracture union. Only one (3.2%) of the 31 HR cases had a dislocation. The sub-analysis of the different osteosynthesis methods showed a higher incidence of excessive sliding and the nonunion of the fracture fragment in the DHS group than that in the CMN group (*p* = 0.004 and *p* = 0.022, respectively). The different surgical methods did not significantly vary in other outcome variables, such as the re-operation rate, mortality, and hip function. Conclusions: For the surgical treatment of combined femoral neck and trochanteric fractures, osteosynthesis did not differ significantly from an HR in terms of the overall postoperative complications, reoperation and mortality rate, and hip function, however, the risk of nonunion and more mechanical complications should be considered when choosing a DHS. Our suggestion for the treatment of a femoral neck and ipsilateral trochanteric fracture is that a surgeon should choose wisely between an HR and a CMN depending on the patient’s age, the displacement of the femoral neck, and one’s expertise.

## 1. Introduction

The incidence of a proximal femoral facture (PFF) is rigorously rising along with the socioeconomic burden due to the global ageing population [1]. A femoral neck fracture (FNF) and intertrochanteric fracture (ITF) are the two most common types of PFF. The clinical significance and interests in PFFs are remarkable due to detrimental postoperative complications, mortality, and a functional decline in polymorbid patients [1]. PFFs are broadly categorized into intracapsular and extracapsular fractures based on patho-anatomy and physiology for therapeutic guidance [2]. Since extracapsular fractures have an undisrupted blood supply, the preservation of the femoral head via osteosynthesis is the gold standard, however, the disruption of the terminal vascularization of the femoral head in intracapsular fractures results in a higher risk of nonunion and avascular necrosis, requiring a surgeon with more discretion in decision-making between osteosynthesis and a hip replacement (HR) [2].

A fracture such as a concomitant ipsilateral femoral neck and intertrochanteric fracture (CIFNITF), which involves both intracapsular and extracapsular fracture components, is difficult to categorize and raises a dilemma of management [3,4,5,6,7,8,9,10,11,12,13,14,15,16,17,18,19,20,21,22,23,24,25,26,27,28,29]. Traditionally, the CIFNITF has been considered an extremely rare type of trauma that displays a bimodal distribution in age and respective injury mechanisms [5,10,20,23]. However, recent literatures suggest otherwise [13,26,27,29]. In line with the recent literatures, the authors believe that the incidence of the CIFNITF is revving up from the expansion of an elderly population and the implementation of personal mobility vehicles. Regardless, the CIFNITF poses management challenges regarding the optimal implants due to the lack of a classification system and an evidence-based consensus. This study surveys the surgical outcomes of different implants in order to assist in selecting the best possible treatment option for the CIFNITF.

## 2. Materials and Methods

### 2.1. Patient Selection

After receiving approval by the institutional review board, we reviewed 2098 patients who underwent surgeries under the diagnosis of FNF and/or ITF between January 2010 and December 2019 in our institution. The inclusion criteria were (1) a diagnosis of CIFNITF confirmed by 3D computed tomography (CT) images, (2) a minimum follow-up period of 12 months postoperatively, and (3) patients aged >18 years. A CIFNITF was diagnosed when (1) more than 50% of the intertrochanteric fracture line was included in the intracapsular pocket at the cross-lined axial and coronal CT view where the femoral head is the largest, or (2) the additional separate fracture center is identified in the transcervical or subcapital region. The presence of a capsular sign was also screened to avoid any misdiagnosis [30]. All diagnoses and surgeries were conducted by two experienced senior orthopedic surgeons who specialized in hip and pelvic trauma for over 20 years. A total of 115 patients were included in the study (Figure 1).

### 2.2. Surgical Technique

We categorized the included patients into three groups according to the following surgical methods: (1) a cephalomedullary nail (CMN), (2) a dynamic hip screw (DHS), and (3) an HR. The compression hip nail (TDM, Seongnam, Republic of Korea) and proximal femoral nail (DePuy Synthes, Solothurn, Switzerland) were used for the CMN, and the DHS system (Jeil Medical, Seoul, Republic of Korea) for the DHS surgery. For the DHS surgery, a trochanteric stabilizing plate (TSP) was applied in all cases. In all osteosynthesis surgeries, we achieved an extramedullary or neutral reduction in either the anteroposterior (AP) or lateral view of a simple radiograph and an acceptable lag screw position [31,32].

For the HR surgery, we performed a total hip arthroplasty using the Bencox^®^ Cup System (Corentec, Seoul, Republic of Korea) with the Bencox^®^ stem (Corentec, Seoul, Republic of Korea) for patients aged <70 years and bipolar hemiarthroplasty surgery using the C2 stem (Lima-LTO, Udine, Italy) or Bencox^®^ stem (Corentec, Seoul, Republic of Korea) for the femoral component in patients aged >70 years. We applied trochanter reinforcement procedures with either a tension band wiring or trochanteric grip plate, only when an unstable abductor or trochanteric fragment was noticed intraoperatively. Figure 2 shows example cases of each treatment modality.

### 2.3. Data Collection

The demographic data and perioperative surgical variables, including the patient’s age, sex, T-score of the bone mineral density (BMD), operative time, type of anesthesia during surgery, and the duration of their hospital stay were collected. The postoperative complications were analyzed for the outcome measure. In osteosynthesis surgeries, the incidence of osteonecrosis of the femoral head (ONFH), cut-out of lag screw, cut-through of lag screw, excessive sliding, and the nonunion rate were investigated. An excessive sliding was defined as a lag screw sliding > 15 mm [33]. We defined a nonunion, as no radiologic evidence of a callus or bridge formation, at least 15 weeks after surgery either AP or in the lateral view of simple radiographs [34].

In an HR, we investigated the incidence of a dislocation and implant loosening during the follow-up period. As overall complications in all surgeries, we investigated the incidence of a heterotopic ossification, venous thromboembolism (VTE), and deep infection. We also studied the revision rate in both osteosynthesis and HR surgeries and determined the mortality rate during the hospitalization period and within 12 months of the postoperative period. For the outcome measurement, we collected the data for the Harris hip score (HHS). All of the radiological parameters were measured by two forementioned orthopedic surgeons, and any disagreements were resolved through discussion between the investigators.

### 2.4. Statistical Analysis

The baseline characteristics were summarized using standard descriptive statistics. The continuous variables were described as the mean (standard deviation) or median (range). A Kruskal–Wallis test was utilized for the comparison of three types of surgeries, and the independent T-test or Mann–Whitney U test were applied for direct one-to-one comparisons of each type of surgery (e.g., CMN vs. DHS). The categorical data were evaluated using the chi-square test or Fisher’s exact test, as applicable, and were summarized as the absolute frequency and percentages. All statistical analyses were performed using PASW Statistics version 18.0 (IBM Corp., Armonk, NY, USA). A statistical significance was defined as a *p*-value < 0.05. Additionally, we evaluated the trends of a CIFNIF incidence in our hospital.

## 3. Results

### 3.1. Patient Demographics and Perioperative Details

The overall incidence of CIFNITF was 6.1% (*n* = 115). The median (range) age of the included patients was 79 (40 to 95) years, and no significant differences were found among the three groups. Seventy-three of the 115 patients were female (63.5%), and females were predominant in all three groups. The median (range) T-score of BMD was −3.3 (−7.2 to 0.8). The CMN had the shortest mean operative time (standard deviation) of 85.6 min (31.1), followed by 94.7 min (22.3) during the DHS, and 107.3 min (37.2) during the HR (*p* = 0.021). When compared to the different types of surgery, significant differences in the operative time were noted between the CMN and HR (*p* = 0.003), and between the DHS and HR (*p* = 0.025). The patient demographics and perioperative details such as the types of anesthesia and the length of the hospital stay did not vary significantly. The basic demographic and perioperative details are elaborated in Table 1.

### 3.2. Postoperative Complications, Mortality Rates and Outcome

We compared the incidence of ONFH, lag screw cut-out, cut-through, excessive sliding, and nonunion of the fracture between the CMN and DHS groups. Seven of the 68 patients (10.3%) in the CMN and 2 of 16 (12.5%) in the DHS were diagnosed with post-traumatic ONFH, with no significant difference between the two groups. There were two cases of cut-out, one case of cut-through in the CMN, and none in the DHS (*p* = 1.000 on both the cut-out and cut-through). The incidence of excessive sliding was significantly higher in the DHS than that in the CMN (25% and 1.5%, respectively [*p* = 0.004]). Four of 16 (25%) patients in the DHS and 3 of 68 (4.4%) patients in the CMN had a nonunion of fracture, and the difference was significant (*p* = 0.022). In the HR, there was one case of dislocation, and no implant loosening was observed. The overall complication rates of heterotopic ossification, VTE, deep infection, and revision surgery displayed no significant differences among the different surgical options. The overall in-hospital mortality rate was 5 of 115 (4.3%), and the one-year mortality rate was 13.9% (16 of 115). The HHS could only be collected from 24 patients in the CMN, 4 in the DHS, and 28 in the HR. The median (range) HHSs were 89.5 points (65 to 95), 78.5 points (65 to 91) in CMN, 91.0 points (85 to 95) in the DHS, and 88.5 points (85 to 95) in the HR, respectively (*p* = 0.819). The details of the surgical outcomes are listed in Table 2.

### 3.3. Trend in the Incidence of CIFNITF

The annual trends in the incidence of CIFNITFs and PFFs were investigated. The bar graph in Figure 3 indicates that the incidence of CIFNITFs randomly varied from 4.1% to 7.0%, while the incidence of PFFs showed increasing trend over time.

## 4. Discussion

The CIFNITF is a challenging type of injury that requires proper surgical planning. However, the underestimation of the incidence of this injury has misguided surgeons in diagnosis and in reaching a consensus in both the proper diagnosis and the optimal treatment modality due to the lack of evidence in the literature. To our knowledge, our work is the first large retrospective study that analyzed the postoperative outcomes of different treatment options for a CIFNITF. Our main finding is that the CMN and DHS did not differ significantly from the HR in terms of the postoperative complications, revision rate, in-hospital and one-year mortality rate, and hip function.

We performed a literature review on a CIFNITF and found 24 case reports and 4 small retrospective studies with 128 patients in total (Appendix A) [3,4,5,6,7,8,9,10,11,12,13,14,15,16,17,18,19,20,21,22,23,24,25,26,27,28,29]. The age distribution and injury mechanism showed bimodal characteristics similar to our study, excluding the four retrospective studies that reported a median age with an interquartile range; the median (range) of the age in all cases was 74 (26 to 97) years. The majority of the injury mechanism (*n* = 104) was low-energy trauma, such as a simple fall. The sex ratio was also in concordance with our study as there were 81 females (63.3%). The studies reported before 2011 mostly used plain radiography for the diagnosis. Two studies made diagnosis intraoperatively by fluoroscopy and one study did not realize the injury until several months after the surgery, which suggests that plain radiography may not be sufficient to diagnose a CIFNITF [4,6,7,8,12,13,14,15,17,18,21,25]. The studies after 2011 employed CT as a final diagnostic modality, and the emergence of larger case series since then indicates that a CT-based diagnosis has enabled surgeons to fully assess and correctly diagnose concomitant injuries.

However, even with a CT-based diagnosis, most studies varied greatly in the description of the fracture pattern and in the implant choices due to the lack of a classification system. Yoo et al. categorized a CIFNITF into three-part and four-part fractures, but could not relate the category with treatment choice since they performed a CMN in all cases [29]. Tong et al. suggested a new classification system which incorporated different morphological features of a CIFNITF [26]. However, after reviewing the fracture patterns and treatment results in the literatures and in our study, we came to the conclusion that the most common fracture configuration is the reverse oblique ITF (AO Foundation/Orthopaedic Trauma Association (AO/OTA) fracture classification 31A3) with an additional fracture center at the intracapsular region (122/128 cases in the literatures and 109/115 in our study) [2]. The second most common was the segmental pattern in which a separate intracapsular fracture center ran parallel to the extracapsular fracture line (5/128 in the literature and 4 in our study). The least common was a continuous fracture line of the FNF that extended like a beak down to the lower end of the intertrochanteric ridge or below the lesser trochanteric region (one in the literature and two in our study). Based on the common morphological patterns reported in the literature and in our study, we simplified the CIFNITF classification in Figure 4.

In the literature review, the utilized implants are as follows: 14 HRs, 2 screws, 2 dynamic condylar screws, 10 plate and screw constructs, 45 DHSs, and 55 CMNs (Appendix A). Only one complication of a dislocation was reported after an HR. There were 7 nonunion or mechanical complications out of 55 CMNs and 13 nonunion, mechanical complication, or ONFH out of 45 DHSs. Similarly, we found a higher rate of excessive sliding and nonunion in the DHS than in the CMN. First, the presence of an intracapsular fracture center makes the fixation of these unstable fractures extremely challenging. The high percentage of geriatric patients with osteoporosis also makes it very challenging to operate on the pathological bone [2]. This is clinically significant since the incidence of complex type of PFFs such as CIFNITF will increase along with the rising of geriatric fragility fractures [1]. A correct diagnosis and adequate treatment are key to achieving a better outcome and minimizing the risk of complications.

Unfortunately, since evidence on the treatment of a CIFNITF is weak, we suggest a provisional management algorithm based on the most appropriate surgical option of each fracture entity (Figure 5). In general, a non-displaced FNF and those of a biologically young age are candidates for a DHS, while a hip replacement (HR) is the gold standard for displaced fractures and biologically old patients [2,35,36,37]. Simultaneously, the most feasible fixation method for unstable ITF was weighed [2,38,39]. Although a CMN, DHS, and HR can all be considered for unstable ITF, a CMN should be the treatment of choice for younger patients, while an HR should be the gold standard for older patients when the injury is complicated by an FNF. Based on the literature review and our study results, a DHS seems to carry more burdens in terms of mechanical complications and nonunion. To date, there have been controversies regarding whether a CMN is superior to DHS in the treatment of intertrochanteric fractures [40,41,42]; however, in cases of unstable intertrochanteric fractures, there is a consensus that a DHS surgery is challenging because of the loss of medial buttress and a lateral wall fracture [43,44]. In fact, Kyle et al. has suggested that surgeons refrain from using DHS and consider an HR even for younger patients for managing a CIFNITF [13]. Thus, we recommend that surgeons should carefully select a DHS over a CMN when fixing a CIFNITF, considering the potential risk factors of a high nonunion and excessive displacement rate in DHS than in CMN.

As for the mortality rate in the current study, the overall in-patient mortality rate was 4.3% (5 of 115), and the overall mortality rate at the one-year postoperative follow-up was 13.9% (16 of 115). There were no significant differences among the different surgery groups; the mortality rate ranged from 0 to 6.5% during the inpatient period and from 6.3 to 22.6% during the one-year follow-up period. These data are comparable with the recent data collected from the national registries of several Asian countries from 2013 to 2017 with a review of 10 articles, which showed an inpatient mortality rate ranging from 3% to 7% and a mean 1-year mortality of 17.89% [45].

We initially hypothesized that the incidence of a combined fracture would increase over time, suspecting that this could be affected by an increased life expectancy or medication history of osteoporosis, however, the percentage of combined fractures randomly varied from 4.1% to 7.0% between 2010 and 2019, with the lowest in 2015 and the highest in 2018. Although the overall incidence of an FNF and ITF increased, there was no evidence of any trends in the incidence of a combined fracture (Figure 3). Nevertheless, the incidence of the combined fracture, which is higher than previous literatures, revealed by a preoperative CT analysis suggests that some of the combined fractures may have been either neglected or overlooked.

The present study has several limitations. First, the study is retrospective and a single center cohort in nature. However, although CIFNITFs are common and most surgeons have at least some experiences with them, there have been limited large studies on this fracture type. Therefore, this study is significant in that it presents the largest sample size to date, along with a surgical outcome analysis based on the different treatment options, thereby providing a rational guide to selecting the treatment modality for this combined fracture. Second, we could not fully compare the patient-reported outcome measures as only part of the data could be extracted because of the retrospective nature of the study, and the study population consisted primarily of old age and severely injured patients. We believe a further large prospective study with more detailed outcome variables will provide surgeons with more rational guidance to approaching treatment of CIFNITFs.

## 5. Conclusions

For the surgical treatment of CIFNITFs, our results indicated that osteosynthesis did not differ significantly from an HR in terms of the overall postoperative complications, reoperation and mortality rate, and hip function. However, the risk of nonunion and more mechanical complications should be considered when choosing a DHS. Our suggestion for the treatment of a CIFNITF is that a surgeon should choose wisely between an HR and a CMN depending on the patient’s age, displacement of the femoral neck, and one’s expertise.

## Figures and Tables

**Figure 1 jpm-12-01908-f001:**
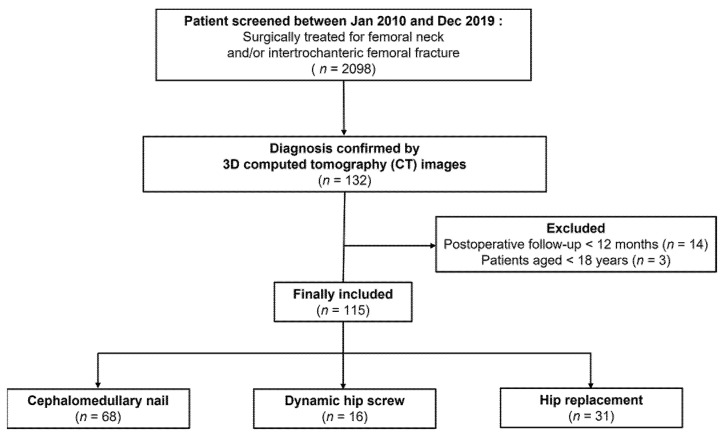
Flow chart of the included patients.

**Figure 2 jpm-12-01908-f002:**
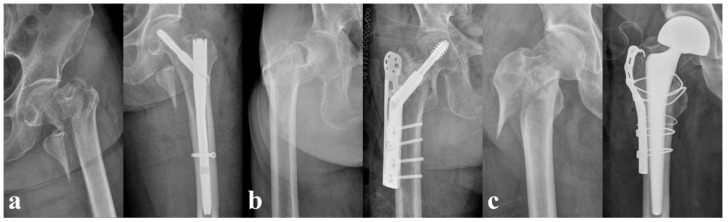
Pre- and postoperative plain radiographs of different treatments. (**a**) A 72-year-old female who underwent cephalomedullary nailing. (**b**) An 88-year-old female who underwent dynamic hip screw fixation. (**c**) A 73-year-old male who underwent bipolar hemiarthroplasty.

**Figure 3 jpm-12-01908-f003:**
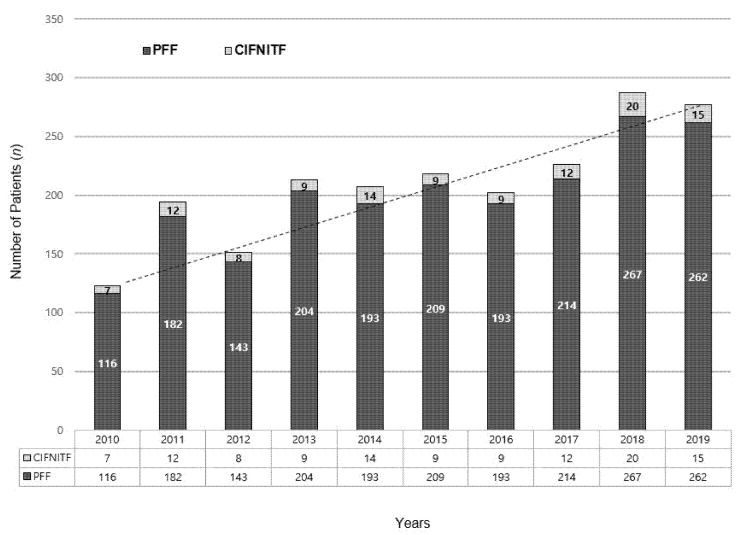
Annual trends of concomitant ipsilateral femoral neck and intertrochanter fracture. (PFF: proximal femoral fracture; CIFNITF: concomitant ipsilateral femoral neck and intertrochanter fracture).

**Figure 4 jpm-12-01908-f004:**
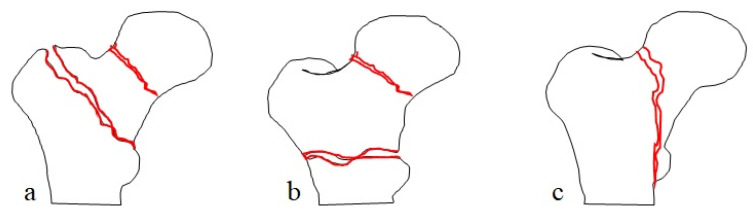
Classification of concomitant ipsilateral femoral neck and intertrochanter fracture. (**a**) Type I: a separate intracapsular fracture center ran parallel to extracapsular fracture line. (**b**) Reverse oblique intertrochanteric fracture with an additional fracture center at intracapsular region. (**c**) A continuous fracture line of femoral neck fracture that extended like a beak down to the lower end of intertrochanteric ridge or below lesser trochanteric region.

**Figure 5 jpm-12-01908-f005:**
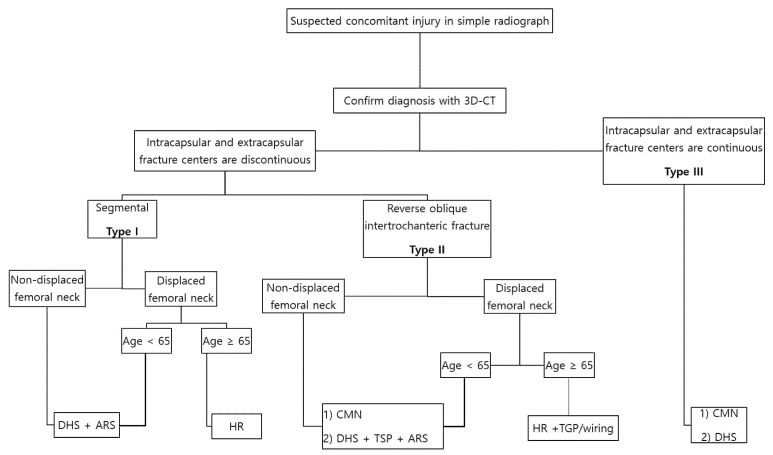
Algorithm for the management of concomitant ipsilateral femoral neck and intertrochanter fracture. ARS: anti-rotation screw; TGP: trochanteric grip plate.

**Table 1 jpm-12-01908-t001:** Patient demographics and perioperative characteristics.

	All (*n* = 115)	CMN (*n* = 68)	DHS (*n* = 16)	HR (*n* = 31)	*p* Value			
					CMN vs. DHS	CMN vs. HR	DHS vs. HR	Overall
**Age**	79 (40 to 95)	47 (40 to 90)	79.5 (42 to 94)	81 (56 to 95)	0.669	0.125	0.574	0.338
**Female**	73 (63.5%)	39 (57.4%)	9 (56.3%)	25 (80.6%)	0.936	0.025	0.096	0.075
**BMD** (T-score)	−3.3 (−7.2 to 0.8)	−3.2 (−4.9 to 0.8)	−3.9 (−4.6 to −3.1)	−3.9 (−7.2 to −0.5)	0.112	0.197	0.881	0.077
**Op time (min)**	92.7 (33.0)	85.6 (31.1)	94.7 (22.3)	107.3 (37.2)	0.174	0.003	0.025	0.021
**Anesthesia ^a^**	53 (46.1%)	36 (52.9%)	6 (37.5%)	11 (35.5%)	0.266	0.107	0.892	0.236
**LOS (day)**	14 (3 to 125)	14 (3 to 119)	16 (6 to 55)	15 (7 to 125)	0.249	0.280	0.991	0.250

Values are expressed as median (range), mean (SD), or *n* (%). ^a^ Type of anesthesia expressed in number of general anesthesia; SD: standard deviation; BMD: bone mineral density; CMN: cephalomedullary nail; DHS: dynamic hip screw; HR: hip replacement; LOS: length of stay; Op: operation.

**Table 2 jpm-12-01908-t002:** Comparison of postoperative outcomes among different treatment modalities.

	All Patients (*n* = 115)	CMN (*n* = 68)	DHS (*n* = 16)	Arthroplasty (*n* = 31)	*p* Value
**Osteosynthesis**					
ONFH		7 (10.3%)	2 (12.5%)	N/A	0.679
Cut-out		2 (2.9%)	0 (0%)	N/A	1.000
Cut-through		1 (1.5%)	0 (0%)	N/A	1.000
Excessive sliding		1 (1.5%)	4 (25.0%)	N/A	0.004
Nonunion		3 (4.4%)	4 (25.0%)	N/A	0.022
**Arthroplasty**					
Dislocation		N/A	N/A	1 (3.2%)	N/A
Implant loosening		N/A	N/A	0 (0%)	N/A
**Total**					
HO	5 (4.3%)	4 (5.9%)	0 (0%)	1 (3.2%)	1.000
VTE	4 (3.5%)	4 (5.9%)	0 (0%)	0 (0%)	0.459
Infection	6 (5.2%)	2 (2.9%)	1 (6.3%)	3 (9.7%)	0.269
**Revision surgery**	6 (5.2%)	4 (5.9%)	1 (6.3%)	1 (3.2%)	1.000
**In-hospital mortality**	5 (4.3%)	3 (4.4%)	0 (0%)	2 (6.5%)	0.674
**1-year mortality**	16 (13.9%)	8 (11.8%)	1 (6.3%)	7 (22.6%)	0.293
**HHS at final f/u**	84.2 (65–95)	73.4 (65–91)	89.5 (85–95)	87.2 (85–95)	0.819

Values are expressed as median (range), mean (SD), or n (%). CMN: cephalomedullary nail; DHS: dynamic hip screw; HR: hip replacement; N/A: not available; f/u; follow-up; HHS: Harris hip score; HO: heterotopic ossification; ONFH: osteonecrosis of femoral head; VTE: venous thromboembolism.

## Data Availability

The data presented in this study are available on request from the corresponding author. The data are not publicly available due to conditions of the ethics committee of our university.

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
