# Peer review of "What Is the Best Treatment Choice for Concomitant Ipsilateral Femoral Neck and Intertrochanteric Fracture? A Retrospective Comparative Analysis of 115 Consecutive Patients"

_jpm, 2022, doi:10.3390/jpm12111908_

Round 1

Reviewer 1 Report

The manuscript “What is the best treatment choice for concomitant ipsilateral femoral neck and intertrochanteric fracture? A retrospective comparative analysis of 115 consecutive patients” by  Han Soul Kim et al. studied which treatment modality is the best for combined femoral neck and intertrochanteric fracture.

Below are my comments and remarks regarding the manuscript:

1. Table 1, please specify the median values - the tests used are non-parametric,

2. Table 1 whether the groups were statistically compared with each other, e.g. CMN vs DHS etc.

3. Table 2. What statistical tests were used to compare the 2 groups?

4. A limitation of the study is retrospective, with limited credibility and lack of data, the conclusions should be slightly more conservative

5. There is no broader discussion of the topic and no discussion of the impact of the factors listed in Table 1 on possible complications and multivariate analysis.

Reviewer 2 Report

This is a study that retrospectively tries to highlight some key outcomes comparing three different surgical options for a rare occurrence of fracture type.

Title

I would suggest removing the question from the title, and rephrasing to something similar to "Treatment choice for concomitant ipsilateral 2 femoral neck and intertrochanteric fracture. A retrospective 3 comparative analysis of 115 consecutive patients"

This would both reduce the length of a title, compressing the keywords and also being more scientifically sound.

Manuscript

By being a retrospective analysis, the sample of patient could have definitely been higher. This is correctly listed as a main limitation of the study.

A topic of discussion should highlight the possible results bias of operating a pathologic bone fracture (osteoporosis and other conditions)

In figure 1, "cephalomedullary nail" word is missing an "a"- please correct in the figure.

Please consider adding an abbreviation for the word "arthroplasty", as you did for DHS, CMN. Maybe use THA, or HR. Also, change this in the entire manuscript.

62-63, 82-83-84 - these rows contain the same information. Please address this only once.

Please always position citations and the end of a sentence, before the dot. Bad example: text text text. [1-2]; Good example: text text text [1-2].

in Table 1 and 2. Can you please clarify what you compared for getting the pvalue listed there?

JPM is not a journal that has orthopedics as a main specialty, therefore, please specific what Fx means in Figure 3.

The conclusion section of this study should be improved as there were many important findings.

More than 50% of references are too old for a state-of-the art principle. Please update the bibliography.

Round 2

Reviewer 1 Report

I have no more comments.

Reviewer 2 Report

Modifications are all made.